# All-Ceramic Passive Wireless Temperature Sensor Realized by Tin-Doped Indium Oxide (ITO) Electrodes for Harsh Environment Applications

**DOI:** 10.3390/s22062165

**Published:** 2022-03-10

**Authors:** Kavin Sivaneri Varadharajan Idhaiam, Joshua A. Caswell, Peter D. Pozo, Katarzyna Sabolsky, Konstantinos A. Sierros, Daryl S. Reynolds, Edward M. Sabolsky

**Affiliations:** 1Department of Mechanical and Aerospace Engineering, West Virginia University, Morgantown, WV 26506, USA; kv0001@mix.wvu.edu (K.S.V.I.); jac0119@mix.wvu.edu (J.A.C.); kathy.sabolsky@mail.wvu.edu (K.S.); kostas.sierros@mail.wvu.edu (K.A.S.); 2Lane Department of Computer Science and Electrical Engineering, West Virginia University, Morgantown, WV 26506, USA; pcd0001@mix.wvu.edu (P.D.P.); daryl.reynolds@mail.wvu.edu (D.S.R.)

**Keywords:** passive wireless sensor, high temperature application, LC resonator, metamaterial, RFID, indium tin oxide, electroconductive ceramic sensor, solid-state sensor, ceramic capacitor

## Abstract

In this work, an all-ceramic passive wireless inductor–capacitor (LC) resonator was presented for stable temperature sensing up to 1200 °C in air. Instead of using conventional metallic electrodes, the LC resonators are modeled and fabricated with thermally stable and highly electroconductive ceramic oxide. The LC resonator was modeled in ANSYS HFSS to operate in a low-frequency region (50 MHz) within 50 × 50 mm geometry using the actual material properties of the circuit elements. The LC resonator was composed of a parallel plate capacitor coupled with a planar inductor deposited on an Al_2_O_3_ substrate using screen-printing, and the ceramic pattern was sintered at 1250 °C for 4 h in an ambient atmosphere. The sensitivity (average change in resonant frequency with respect to temperature) from 200–1200 °C was ~170 kHz/°C. The temperature-dependent electrical conductivity of the tin-doped indium oxide (ITO, 10% SnO_2_ doping) on the quality factor showed an increase of *Q_f_* from 36 to 43 between 200 °C and 1200 °C. The proposed ITO electrodes displayed improved sensitivity and quality factor at elevated temperatures, proving them to be an excellent candidate for temperature sensing in harsh environments. The microstructural analysis of the co-sintered LC resonator was performed using a scanning electron microscope (SEM) which showed that there are no cross-sectional and topographical defects after several thermal treatments.

## 1. Introduction

Real-time temperature awareness is vital for every industrial process, and the temperature sensor is one of the most widely used sensors in nearly all industries. Based on the working temperature region, temperature sensors can be classified as low to moderate (25–300 °C), moderate to high (300–1200 °C), and very high to ultra-high (1200–2000 °C) [1]. Thermometers and optical sensors (infrared sensors) are used in sensing applications for low-temperature regions (−250 to 300 °C). Conventional resistance temperature detectors (RTDs) and thermocouples are extensively used in moderate (500 °C) to ultra-high temperature regions (2000 °C) [2]. Thermocouples have limited applications where a cold and hot junction should be maintained for accurate temperature measurement. This leads to numerous challenges during interconnecting the sensors with sophisticated furnaces, gasifiers, reactors, and nonstationary parts (such as turbine blades and rotors) [2,3]. Additionally, conventional temperature sensors have been predominantly fabricated with refractory metals and metal alloys, such as Ni, Cr, Si, Mg, Pt, and Rh, which are prone to oxidation/corrosion and lead to the necessary frequency calibration or replacement [3]. Therefore, a better material system is needed for sensing temperature in ambient air, oxidizing, and corrosive atmospheres to prevent degradation and increase the longevity of the sensor. Alternate technologies have been in development to mitigate the interconnect issue and increase the longevity of the sensors, which include the implementation of wireless sensor architectures [4,5,6,7]. The wireless sensor technology is considered a paradigm shift in many sensing technologies due to their wide range of applications, including temperature realization [2,6,8,9,10,11,12,13,14,15,16,17,18,19,20,21,22]. The conventional material systems, based on metal interconnects and silicon-based semiconductor components, cannot operate at elevated temperatures due to poor thermal and chemical stability. Additionally, the conventional wireless sensor requires a power source for its operation, which will be impractical in harsh environmental conditions. Therefore, researchers have been investigating high-temperature material systems for different types of passive wireless sensing technology based on their working principle.

Wireless sensors are classified into several types based on their working mechanism; however, in this work, the discussion is narrowed down to passive wireless devices because they do not need an external power source for their operation [2,5,10,15,16,20,21,23,24,25,26,27,28,29]. A passive wireless sensor is defined as a device that does not require an external power source for its operation and is fabricated using passive electrical components. Among the passive wireless sensing technologies, surface acoustic wave (SAW) and wireless LC resonators were evaluated for temperature sensing in harsh environmental conditions [1,30,31,32,33]. Due to the poor stability of the materials used in the fabrication of SAW sensors, they are limited to a temperature range below 600 °C [1,30,31,32]. Even though SAW sensors provide wireless interrogation over long range (>50 cm), the surface acoustic wave carrying the temperature information can be influenced by the operating environment, geometry, and any parasitic noise in the path of the acoustic waves [31,32]. Hence, SAW sensors may not be a good choice for real-world applications where the operating environment is not ideal as presented at lab-scale testing. Additionally, the piezoelectric crystals used to fabricate the SAW sensors, such as langasites, are limited to operate below 600 °C [31]. The LC resonator operation is based on the mutual inductive coupling between the sensor and an interrogator antenna. The dielectric permittivity change with respect to temperature is the working principle of most LC resonators. The dielectric material can be made with various refractory ceramics that are stable at high temperatures. This would also provide several degrees of freedom to interrogate the wireless response [13,34,35]. Due to the versatility and ease of fabrication of the LC resonator, research has focused on developing sensors based on the principle of LC resonance [16,17,20,21,24,27,28,34,36,37]. Wang et al. first reported a temperature sensor based on the principle of LC resonance that works up to 235 °C [36]. Later, a few researchers developed a passive wireless temperature sensor up to 1000 °C using low-temperature co-fired ceramic (LTCC refers to co-sintering of the metallic electrodes printed onto a green dielectric ceramic tape) materials, where they had similar sensor architectures [19,24,28]. The reported LC resonators were fabricated by simply depositing various metallic electrodes onto dielectric ceramic substrates for temperature sensing up to 1000 °C [16,20,24,27]. Although the LC resonators showed adequate performance up to 1000 °C, the signal loss at temperatures >500 °C is significant due to the exponential increase in resistance of the metallic electrodes [6,9,16,20,24,38]. In addition, most of the metals used in the reports have a relatively low melting points, and the microstructure tends to coarsen and delaminate from the dielectric ceramic substrate at elevated temperatures, leading to failure when they undergo several thermal loading cycles [34]. Ji et al. used Pt electrodes with the same LC resonator architecture to increase the range of temperature sensing up to 1400 °C, but the sensor still suffered signal loss of >1000 °C due to an exponential increase in resistance of Pt [17]. So far, all the reports in the literature have used various metallic electrodes (Ag, Pt, Cu, Ag–Pd alloy) deposited on dielectric ceramic substrates to realize the LC resonator for temperature-sensing applications [16,17,20,21,24,27,28,32,33,34,35,36]. It is evident from their results that the sensor incurs significant signal loss at high temperatures irrespective of the metallic electrodes used in the fabrication process. An alternative material system to replace the metallic electrodes is required to have better sensitivity and low signal loss at elevated temperatures.

In this work, an all-ceramic LC resonator sensor was designed, fabricated, and tested using an electronically conductive ceramic material to replace the conventional metallic electrodes. Tin-doped indium oxide (ITO) with 10% tin oxide (SnO_2_) doping concentration was chosen as the electronically conductive ceramic material to replace the metallic electrodes due to its high electrical conductivity with excellent high-temperature stability [35,36,37,38,39,40,41,42,43]. The geometry and the physical attributes of the LC resonator were modeled in ANSYS HFSS using the actual electrical properties of the material system used in the fabrication process. Additionally, the resonator was designed to operate at low power (<0.5 W) to minimize the requirement of sophisticated high-power electronics. The low operating power of the LC resonator limits the resonance frequency below 200 MHz which eliminates parasitic noise/influence from the furnace/reactors since the heating elements operate at higher frequencies. Ideally, the resonant frequency shifts with respect to both capacitance and inductance. Inductance is only affected by the change in the geometrical dimension of the planar inductor. The thermal expansion coefficient of the ITO electrode, which makes up the planar inductor coil and the Al_2_O_3_ substrate, is on the order of 10^−6^/°C, which has a negligible influence on straining the planar inductor. Therefore, the inductance does not significantly alter the resonant frequency, where the inductor part of the equivalent circuit is used only as a transmitting antenna. Among the several parameters that influence the capacitance, only the dielectric permittivity is directly proportional to temperature. As Chen et al. demonstrated in their work, the dielectric properties of polycrystalline Al_2_O_3_ as a function of temperature, frequency, and impurity content showed that the permittivity of polycrystalline Al_2_O_3_ with 99% and 96% purity strongly depends on the temperature at frequencies >500 kHz [44]. Thus, the change in resonant frequency is assumed to be predominantly influenced by the change in temperature. In this work, the effect of capacitance and inductance as a function of temperature was individually studied to validate the mechanism. The wireless response of the LC resonator was analyzed during the heating and cooling cycle to understand the thermal cycles of the wireless response. In addition to the wireless response evaluation of the LC resonator, the microstructure of the LC resonator that underwent multiple thermal cycles was characterized. The microstructures of the LC resonator deposited on the Al_2_O_3_ substrate were evaluated using a scanning electron microscope (SEM) to corroborate the stability of the wireless response at high temperatures.

## 2. Materials and Methods

### 2.1. Temperature Sensing Principle and Fabrication

The temperature sensing of the LC resonator is based on the principle of a change in the resonant frequency as a function of temperature, and the resonant frequency (*f_r_*) is given by
(1)$$f_{r}=\frac{1}{2\pi \sqrt{L_{r}C_{r}}}$$
where *L_r_* and *C_r_* are the inductance and the capacitance of the LC resonator, respectively. Any variations in the inductance or the capacitance will be reflected as a change in the resonant frequency. The shift in the resonant frequency can be wirelessly interrogated with an external antenna inductively coupled with the LC resonator.

The electrical model of the LC resonator was built using the ANSYS HFSS package (Canonsburg, PA, USA). The LC resonator used in this work is composed of a parallel plate capacitor coupled with a planar inductor on an Al_2_O_3_ dielectric substrate. The finite element analysis was performed under practical working conditions with actual material properties such as the dielectric permittivity of the Al_2_O_3_ (9.6) and the electrical conductivity of ITO electrodes (90 S/cm) at room temperature. It is necessary to consider the effect of the quality factor on the sensitivity of the LC resonator since the resolution and the sensitivity depend on the quality factor. The quality factor (*Q_f_*) of the LC resonator is defined as
(2)$$Q_{f}=\frac{1}{R_{r}}\sqrt{\frac{L_{r}}{C_{r}}}\;,\;Q_{f}\propto \;\frac{1}{R}$$
where *R_r_* is the resistance that arises from the planar inductor coil and the electrodes of the capacitor. A high *Q_f_* can be achieved by minimizing the resistance (*R_r_*) and the capacitance (*C_r_*) while maintaining high inductance (*L_r_*). Therefore, the LC resonator was specifically designed to have high inductance, low capacitance, and relatively low resistance. The equation for the parallel plate capacitor (*C_r_*) is written as
(3)$$C_{r}=\frac{\varepsilon_{0}\varepsilon_{r}A}{d}$$
where *ε*_0_ is the permittivity of free space (8.854 × 10^−12^ F/m), *ε_r_* is the relative permittivity of the dielectric (in this case it is Al_2_O_3_ = 9.6), *A* is the cross-sectional area of electrodes, and *d* is the separation distance between the parallel plates. From Equation (2), the desired capacitance was modeled by changing the cross-sectional area and the separation distance (in this case, the thickness of the Al_2_O_3_ dielectric). Figure 1a shows the 3D schematic of the parallel capacitor structure. Table 1 shows the design parameter of the parallel place capacitor and the planar inductor. The area of the circular electrodes was chosen to be ~9.68 mm^2^, and the corresponding diameter is ~3.51 mm. It can be inferred from Figure 1 that the diameter of the dielectric layer was 10% larger than the electrodes to avoid the capacitor short-circuit during the fabrication and sintering process. In order to achieve a capacitance of ~20.56 pF, the thickness of the Al_2_O_3_ dielectric layer was calculated to be 40 µm for the given area.

The square planar inductor was designed by the model presented by Mohan et al. [45]. The expression of such square planar inductor is given by
(4)$$L=\frac{1.27\times \mu_{0}n^{2}d_{avg}}{2}\left[ln\frac{2.07}{\rho }+0.18\times \rho +0.13\times \rho^{2}\right]$$
where *µ*_0_ is permeability of free space (4π × 10^−7^ H/m), *n* is the number of turns of the inductor coil, *d_avg_* is the average diameter of the inductor, and *ρ* is the filling ratio. The *d_avg_* and *ρ* is given by
(5)$$d_{avg}=\frac{d_{in}+d_{out}}{2}$$
(6)$$\rho =\frac{d_{out}-d_{in}}{d_{out}+d_{in}}$$
where *d_in_* and *d_out_* are the inner and the outer diameter of the square planar inductor, respectively. Equation (4) determines the inductance based only on the geometrical parameters, assuming the electrodes are perfect metallic conductors, but in reality, the resistance of the ITO is embedded within the planar inductor. When the current flows within the spiral inductor, it creates a magnetic field with field strength (*B*), given by
(7)$$B=\frac{\mu_{0}I}{2\pi r}$$
where *µ*_0_ is the permeability of free space (4π × 10^−7^ H/m), *I* is the current flowing through the inductor coil, and *r* is the radial distance. Equation (7) shows that the magnetic field strength (*B*) is directly proportional to the current (*I*). The resistance of the electrode will restrict the flow of current, thus attenuating the LC oscillation over a period of time. When the resistance is too high, it will significantly decrease the magnetic field strength and the quality factor of the LC resonator. In order to maintain low inductive resistance, the number of turns (*n*) should be minimum, and the cross-sectional area of the thick film electrodes should be increased. Several iterations of the design were modeled to achieve the correct geometrical form factor of the planar inductor. Based on the design parameters shown in Table 1, a three-turn planar inductor coil with 2 mm width and spacing between the electrodes, and a thickness ranging from 30–40 µm, provided an inductance ranging from 0.81–0.83 μH. The theoretical resistance of the planar inductor is calculated to be ~35–40 ohms, assuming the conductivity of the ITO is 90 S/cm at room temperature. Figure 1b,c show the schematic of the lumped LC resonator with a dimension of 45 × 30 mm. The schematic of the LC resonator simulation setup in ANSYS—HFSS is shown in Appendix A. The theoretical resonant frequency of the LC resonator presented in Figure 1 is 38.65 MHz.

The LC resonator was fabricated using a screen-printing technology with the functional ceramic inks of ITO and Al_2_O_3_ particles. The ITO and Al_2_O_3_ ink were used for the electrical conductor and capacitor dielectric of the LC resonator, respectively. The ITO (In_2_O_3_ with 10% SnO_2_ doping) and Al_2_O_3_ particles were purchased from Beantown Chemicals (Hudson, NH, USA) and Alfa Aesar (Haverhill, MA, USA), respectively. Since the Al_2_O_3_ has an average particle of 1 µm with a uniform distribution, no further milling was required to formulate the functional ceramic ink of the uniform particle size distribution. Knowing the particle size and their distribution is important, because particles >10 µm with a nonuniform distribution will create uneven film thickness and defects during the screen-printing process. Therefore, it is necessary to have particles below 10 µm with a normal distribution. However, Figure 2 shows that the average particle size of the as-purchased ITO ranged from 1–100 µm; therefore, the particles were milled in a high-energy attrition mill for ~4 h in isopropanol to achieve a uniform particle size distribution. It can be inferred from Figure 2 that the milled ITO particles show a uniform distribution centered at 1.26 µm, and there are no particles above 7 µm, which favors the printing and aids in the densification during the sintering process. The ITO and the Al_2_O_3_ functional ceramic inks were prepared by dispersing the particles in an ethyl cellulose–terpineol organic matrix with a drop of fish oil to aid in the dispersion of the ceramic particles.

A 230-mesh screen with the design shown in Figure 3 was purchased from UTZ Technologies (Little Falls, NJ, USA). Figure 3a shows the 2D schematic of the bottom capacitor electrode connected to the planar inductor deposited as the first layer on a 0.5 mm thick Al_2_O_3_ substrate, and the film was completely dried before depositing the next layer. The same process was followed to deposit the Al_2_O_3_ dielectric layer (Figure 3b) and the top ITO electrode layer (Figure 3c) to complete the LC resonator circuit. The screen-printed LC resonator was co-sintered at 1250 °C in a muffle furnace under ambient atmosphere for 4 h with a heating rate of 2 °C/min and cooled at 5 °C/min. An optical microscope was used to investigate macroscopic defects during the printing and after the sintering process. Figure 3e shows the optical microscope image of the co-sintered LC resonator on an Al_2_O_3_ substrate. The defect-free LC resonators were used for the wireless and microstructural characterization. Individual parallel plate capacitors and inductors were printed separately on an Al_2_O_3_ substrate with the same layer thickness to characterize the capacitance and the inductance as a function of temperature. 

### 2.2. Material and Wireless Characterization

The microstructural characterization of the LC resonator was performed using a scanning electron microscope (SEM, JOEL JSM-7100F, Peabody, MA, USA). The capacitance and inductance as a function of temperature were independently measured using an Agilent E4980A LCR meter (now Keysight, Colorado Springs, CO, USA). As stated previously, the LC resonator was characterized by the principle of mutual inductance. The mutual inductance (*M*) between the antenna and the sensor is expressed as
(8)$$M=k\sqrt{L_{a}L_{r}}$$
where *L_a_* and *L_r_* are the inductance of the antenna and the resonator, respectively; *k* is the coupling coefficient between the inductor and the antenna, and it is given by [12,13],
(9)k=21+232d12rarr232,
where *d*_12_ is the distance between the antenna and the inductor, *r_a_* and *r_r_* is the radius of the antenna and the resonator, respectively. The coupling coefficient determines the strength of the mutual inductance as a function of the distance between mutually coupled inductors. By substituting the physical parameters of the antenna and planar inductor into Equation (9), the maximum theoretical distance between these elements is found to be ~4 cm, above which the mutual inductance is zero (no magnetic flux between the interrogator antenna and the inductor). The interrogator antenna will excite the LC resonator through inductive coupling. The resonant frequency is indirectly measured by analyzing the change in the input impedance (*Z_in_*) at the antenna, which is given by
(10)$$Z_{in}=R_{a}+j\omega Z_{a}+\frac{1}{j\omega C_{a}}+\frac{{\left(\omega M\right)}^{2}}{R_{r}+j\omega L_{r}+\frac{1}{j\omega C_{r}}}=R_{a}+X_{L\left(a\right)}+X_{C\left(a\right)}+\frac{{\left(\omega M\right)}^{2}}{R_{r}+X_{L\left(r\right)}+X_{C\left(r\right)}}$$
where *R_a_*, *X_L(a)_*, and *X_C(a)_* are the resistance, inductive, and capacitive reactance of the antenna, respectively. Similarly, *R_r_*, *X_L(r)_*, and *X_C(r)_* are the resistance, inductive, and capacitive reactance of the LC resonator, respectively. It is evident from Equation (10) that the *Z_in_* is composed of the sensing parameter, *C_(r)_* (capacitance which is proportional to the change in temperature). The impedance (*Z_o_*) at the antenna port is assumed to be 50 Ω (which is a standard in RF technology), where the change in the input impedance (*Z_in_*) from the LC resonator is detected by the antenna theory relation given by
(11)$$S_{11}=\frac{Z_{in}-Z_{o}}{Z_{in}+Z_{o}}$$

Since the maximum interrogation distance can reach only up to 4 cm, the conventional copper antenna was replaced with a single loop Pt wire to interrogate at high temperature. The diameter of the Pt wire was 100 mil (~2.45 mm), and the diameter of the single loop antenna was 30 mm, which is the outer diameter of the LC resonator. The antenna was embedded within the Al_2_O_3_ ceramic insulation to further protect it from the hot zone because the increase in resistivity due to temperature will attenuate the mutual inductive coupling. The fabricated LC resonator was characterized from 25–1250 °C in an MTI Corp muffle furnace (Richmond, CA, USA) at ambient atmosphere with an in-situ data acquisition using a Fieldfox N91123A vector network analyzer (VNA, Keysight, Santa Rosa, CA, USA). The VNA swept the frequency from 1 to 100 MHz with a 10 kHz/sec sweep rate to characterize the resonant frequency of the LC resonator.

## 3. Results and Discussion

### 3.1. Microstructural Analysis of the LC Resonator

The microstructural characterization is crucial for understanding the effect of microscopic defects on the wireless response of the LC resonator. Since the multi-layer capacitor structure was co-sintered at 1250 °C, it is vital to analyze the interface of the ITO and Al_2_O_3_ dielectric layer. The interfacial reaction may affect the performance and the resonant frequency of the LC resonator. It should also be noted that the LC resonators went through four thermal cycles (sintering + three thermal cycles for sensor characterization) before analyzing the microstructure. Figure 4 reveals the cross-section of the capacitor where the bright top and the bottom layer are the ITO electrodes, and the dark middle layer represents the Al_2_O_3_ dielectric layer. It is noteworthy to see that the grains are percolated and well sintered across all three layers. The co-sintered ITO–Al_2_O_3_–ITO capacitor structure showed a well-defined interface. As previously discussed, the thickness of the Al_2_O_3_ dielectric determines the capacitance where thickness is found to be 40 ±1 μm, which is within a 2–3% error margin of the theoretical model. Additionally, the thickness of the ITO layers also agrees with the theoretical calculations based on the mesh of the screen and the solid loadings of the ink.

Figure 5a and Figure 5b,c shows the cross-sectional and topography SEM analysis of the planar inductor deposited on the polycrystalline Al_2_O_3_ substrate, respectively. The substrate–ITO interface is denser than the ITO–Al_2_O_3_ interface of the parallel plate capacitor because the planar inductor was printed directly on the 99% dense Al_2_O_3_ substrate, where the particles have a higher packing factor. Additionally, the pressure applied by the squeegee during the screen-printing might have helped the particles to pack denser at the substrate–ITO interface. However, there is no noticeable secondary phase at the substrate–ITO interface, which is similar to the ITO–Al_2_O_3_–ITO interface of the capacitor. The topography shows that the particles are percolated in all three dimensions with an average grain size of 2.2 ± 0.62 μm. The magnified topographical image presented in Figure 5c clearly reveals that the ITO grains are well-sintered, and no grain growth/coarsening was observed, which is detrimental to the sensing capability of the LC resonator.

### 3.2. Electrical Properties of the LC Resonator

The published reports on the inductive coupling-based passive wireless sensor have not individually analyzed the effect of the capacitor and the planar inductor as a function of temperature [1,4,5,9,16,17,19,20,21,24,27]. In an ideal scenario, the LC resonator was modeled assuming only the capacitance changes as a function of temperature. However, at high temperatures, multiple parameters can influence the electrical properties of the circuit elements that make up the LC resonator. Individually analyzing the inductance and capacitance will provide experimental evidence of their influence on the change in the resonant frequency as a function of temperature. Therefore, the capacitance and the inductance were individually analyzed as a function of temperature. As previously discussed in Section 2.1, the increase in electrical conductivity of the ITO electrodes increases the *Q_f_* of both the capacitor and the inductor. The planar inductor was screen-printed on an Al_2_O_3_ substrate and sintered with the sintering profile followed for the LC resonator. Pt electrical leads were made at the contact pads with ITO ink and thermally bonded at 1250 °C for 4 h to analyze the inductance as a function of temperature. The in-situ inductance, resistance, and *Q_f_* were measured using an Agilent E4590 LCR meter. At room temperature, the measured inductance was 0.87 µH, which is 0.05 µH higher than the calculated capacitance due to parasitic inductance from the interconnects. The change in the inductance is ~5 nH as the temperature increased from 25–1200 °C, which will not significantly influence the resonant frequency shift. The increase in the inductance is due to an increase in the electrical conductivity of the ITO as a function of temperature. The increase in the conductivity of the ITO increases the flow of current in the spiral inductor coil, which in turn increases the inductance. For the same reason, the *Q_f_* of the inductor also increased from 35 to 42 as the temperature was raised from 25 to 1200 °C. This clearly shows that the semiconducting nature of the ITO electrodes increases the signal strength of the LC resonator as opposed to the metallic electrodes.

Similarly, the temperature-dependent capacitance of the parallel capacitor was separately analyzed from 100 to 1200 °C at a heating and cooling rate of 2 °C/min. At room temperature, the measured capacitance was 24.27 pF, which is 3.71 pF higher than the calculated capacitance due to parasitic capacitance from the interconnects. As previously discussed, the same fabrication and the test setup were followed to characterize the parallel capacitor. The change in capacitance is recorded using a LABVIEW program at a temperature interval of 1 °C. Figure 6 shows the temperature versus capacitance relation of the parallel capacitor where there is a semi-linear increase in capacitance with a third-order polynomial fit. The capacitance response can be divided into two regions depending on the temperature-dependent dielectric polarization mechanism of the Al_2_O_3_. Binary oxides such as Al_2_O_3_ have variable dielectric permittivity based on the operating temperature, voltage, and frequency. Since the frequency and the voltage was kept constant at 2 MHz and 1 V_AC_ during the characterization, it is assumed that the permittivity changes due to the change in temperature. When the temperature is increased, the quasi-free-charge carriers present within the dielectric crystal are thermally excited and respond to the applied external field. Chen et al. predicted that the quasi-free-charge carriers arise from the dopants or impurities and defects (grain boundaries) present in the Al_2_O_3_. Further raising the temperature (>400 °C), the quasi-free-charge carriers gain more energy and generate a larger polarization at elevated temperatures. Thus, the dielectric permittivity significantly increases at temperatures >400 °C, which drastically increases the capacitance. The average change in capacitance (Δ*C*) is ~26 pF from 100–1200 °C with a unit average of 0.234 pF/°C. The parallel plate capacitor shows a hysteresis error of ~1.39%. During the cooling cycle, the thermal relaxation of the quasi-free-charge carriers is slower in the Al_2_O_3_ crystal and causes a time lag, leading to a capacitive hysteresis. Additionally, the capacitance was measured directly on the substrate, while the thermocouple measures the average temperature change of the furnace. This may create differential temperature recording where the furnace cools slower than the substrate, causing a difference in the temperature and the capacitance measurement.

### 3.3. Modeling and Wireless Characterization of the LC Resonator

Initially, a computational simulation was performed using the ANSYS HFSS software package, where the dielectric permittivity of the Al_2_O_3_ was kept constant at 9.6 to exclusively study the effect of electrical conductivity of the ITO on the wireless response. The LC resonator was placed in an airbox in HFSS with perfect electric (E) and perfect magnetic (H) boundary conditions along the *x*-axis and *z*-axis, respectively. Figure 7a shows that the magnitude increases (the peak narrows) as a function of temperature, proving that the LC resonator has better sensitivity at high temperatures. For better clarity, the conductivity of ITO at four equal temperature intervals (25 °C, 400 °C, 800 °C, and 1200 °C) was used in Figure 7. The signal strength at room temperature (where σ_ITO_ = 90 S/cm) is −10 dB, where the signal strength was increased to −15.8 dB when the temperature reached 1200 °C. It should be noted that there is no observed peak shift at different temperatures because the permittivity was kept constant, which is the parameter responsible for peak shift with respect to temperature. As previously stated, the *f_r_* at room temperature is 38.65 MHz, which can be confirmed by the simulation shown in Figure 7a. To better understand the Δ*f_r_* as a function of temperature, both the dielectric permittivity of the Al_2_O_3_ layer and the conductivity of the ITO electrodes were changed in the HFSS software package. The dielectric permittivity was calculated from the capacitance versus temperature shown in Figure 6 at their respective temperature interval (25 °C, 400 °C, 800 °C, and 1200 °C). Since the Δ*L* from room temperature to 1200 °C is minimal (~5 nH), it was kept constant during the simulation. Figure 7b shows the Δ*f_r_* as a function of temperature, the increase in permittivity shifts with the *f_r_* and is in concordance with the sensing principle.

In this work, the return loss of the antenna (*S*_11_ parameter) was utilized to characterize the resonant frequency shift as a function of temperature. Recalling Equation (11) from Section 2.2, the input impedance of the antenna is composed of the impedance of the LC resonator due to mutual inductive coupling. Therefore, any change in the resonant frequency of the LC resonator can be wirelessly interrogated by the antenna. The LC resonator was initially evaluated at room temperature by varying the interrogator antenna–inductor distance from 1–4 cm to determine the maximum coupling distance without compromising appreciable signal loss. It was found that a working distance of 2.5 cm provides a good trade-off between inductive coupling and signal loss. It must be noted from Figure 8 that a 1-inch (~2.54 cm) thick piece of Al_2_O_3_ fiber insulation was set between the LC resonator and antenna to shield the antenna from high temperature. The results show that this additional thick dielectric layer did not affect or interfere with the signal strength. Figure 8 shows the high-temperature setup and the equivalent circuit for characterizing the wireless response with respect to the temperature where the Pt antenna was parallel to the inductor for efficient inductive coupling. An external thermocouple was placed in proximity to the LC resonator to record the temperature of the furnace. The thermocouple and the VNA were connected to the computer via a DAQ card to record the wireless response of the LC resonator as a function of temperature using a MATLAB script. The heating and the cooling rate of the furnace were kept constant at 1 °C/min. The wireless response was recorded at a temperature interval of 100 °C starting at 200 °C up to 1200 °C during the heating and the cooling cycle.

The experimental *f_r_* of the LC resonator at room temperature was measured to be ~35.43 MHz which is 3.1 MHz lower than the theoretical value. It is common for the *f_r_* to deviate from the theoretical value due to parasitic influence from the electrode, inconsistency of the geometry during the fabrication process, and error during the measurement. An LC-based sensor reported by Ji et al. showed a difference of ~74 MHz, which is quite large due to the high capacitance from their sensor design [17]. However, in this work, the difference between the theoretical model and the experimental result is low due to accurately using the measured high-temperature material properties of the components within the computational modeling and the shorter length of the inductor–capacitor electrode. Figure 9a shows the change in *f_r_* as a function of temperature during the heating cycle. As the temperature increases, the *f_r_* decreases, which is in correlation with Equations (1) and (3). The capacitance increase shifts the frequency to the right side of the spectrum due to the inverse correlation of capacitance and *fr*. It is significant to note that the bandwidth of the resonant frequency does not attenuate as the temperature increases, which is contrary to the LC-resonator-based wireless sensor reported in the literature [1,4,5,9,16,17,19,20,21,24,28]. The bandwidth of the LC resonator is given by
(12)$$BW=\frac{f_{r}}{Q_{f}}=\frac{R_{s}}{2\pi L_{s}},BW\propto R_{s},\frac{1}{L_{s}}$$

According to Equation (12), the BW is proportional to resistance and inversely proportional to inductance. In this case, the change in inductance is disregarded because the inductance variation as a function of temperature is minimal (~5 nH), whereas, as stated previously, the resistance of the ITO electrodes decreases as the temperature increases. The steady decrease in the resistance provides a narrow bandwidth and high *Q_f_* even at high operating temperatures in correlation with the simulation shown in Figure 7. The experimental curve shown in Figure 9 is slightly different from the simulation because the simulation was performed under perfect electric and magnetic boundary conditions. However, in reality, it is impractical to achieve such a pristine working environment. Additionally, the experiment also accounts for the effect of the interrogator antenna, which adds parasitic effects to the wireless response. Nevertheless, the experimental analysis obeys the theoretical simulation with a slight change in the peak shape, which can be attributed to several parasitic noises arising from the environment and the instrument. In addition to the thermal and microstructural stability, the ITO electrode provides an improved signal strength at high temperatures, unlike the metallic electrode. Figure 9b shows the wireless response during the cooling cycle of the LC resonator. The *f_r_* follows a similar trend during the cooling and heating cycle.

In order to understand the dynamic response of the LC resonator during the heating and the cooling cycle, the *f_r_* at each temperature was extrapolated from Figure 9 and plotted as *f_r_* vs. temperature. It is clear from Figure 10 that the response from the cooling cycle does not coincide with the heating cycle of the LC resonator. This trend is similar to the capacitor’s response during the thermal loading cycles. The difference in the *f_r_* during the heating and cooling cycle could be due to the following reasons:As stated in Section 3.2, the relaxation of the quasi-free charges during the cooling cycle is slower than the heating cycle due to the volumetric heat capacity of Al_2_O_3_. Therefore, it loses thermal energy slower than gaining the same amount of thermal energy [44]. Hence, during the cooling cycle, the dielectric permittivity of Al_2_O_3_ relaxation is slower than the heating cycle. It was translated to the *f_r_* as temperature lag during cooling.The temperature difference between the surface of the substrate and the thermocouple can be mistaken as the delay in the LC resonator’s response leading to a difference in *f_r_* values during the heating and the cooling cycle. The *f_r_* measured at the surface of the substrate consists of temperature information localized at the substrate’s surface, whereas the thermocouple records the temperature of the furnace.

A similar trend was seen in the LC-based temperature sensor reported by Tan et al., but the difference in their measurements varies by an average of 0.6 MHz at a temperature above 450 °C [46]. Nevertheless, the experiment was repeatable and demonstrated that the peak fit in Figure 10 is a third-order polynomial fit with an R^2^ of 0.99. The sensitivity of the LC resonator was calculated from the slope of the curve presented in Figure 10. The Δ*f_r_* is ~1.7 MHz between 200 and 1200 °C; therefore, the sensitivity of the of LC resonator was calculated to be ~170 kHz/°C between 200 and 1200 °C.

The quality factor (*Q_f_*) defines the sensitivity of the LC resonator and the ability to differentiate signature peaks at each characterization temperature. Therefore, the temperature-dependent resistance, inductance, and capacitance were substituted into Equation (2) to determine *Q_f_* as a function of temperature. The *Q_f_* at room temperature is calculated to be ~34.5, and it increases to ~43 as the temperature reaches 1200 °C. Similarly, the experimental *Q_f_* was measured by the VNA at each characterization temperature to compare with the theoretical model. Figure 11 shows the comparison of the theoretical model versus the measured *Q_f_* of the LC resonator as a function of temperature. The experimental value agrees very well with the model. According to Equation (2), an increase in inductance can increase the *Q_f_*, but in this case, it has a negligible effect because Δ*L* is only ~5 nH. The increase in *Q_f_* predominantly arises from the increase in conductivity of the ITO as a function of temperature. Idhaiam et al. showed that the conductivity of thick film ITO increases from 90 S/cm to 270 S/cm for temperatures between 200 and 1250 °C [38]. A similar ITO/In_2_O_3_ screen-printed thermocouple was presented by Liu et al.; however, the temperature-dependent electrical conductivity was not reported in the literature [37]. The change in *Q_f_* is similar to the trend presented in the temperature-dependent electrical conductivity of ITO, which corroborates the theory that the increase in *Q_f_* is proportional to the electrical conductivity of ITO. It is significant to note that utilizing semiconducting oxide materials with an electrical conductivity of at least 90 S/cm for the LC resonator sensor at high temperatures increased the sensing response and boosts the signal strength at elevated temperatures, unlike metallic electrodes. For instance, the *Q_f_* of the sensor reported by Tan et al. decreased from 30 to 10 as the temperature increased, whereas the *Q_f_* increased from 34.5 to 43 in this work [46]. This demonstrates that the high conductivity, coupled with the semiconducting nature of the ITO electrodes, is an excellent alternative electrode for temperature sensing at high temperatures.

## 4. Conclusions

In this work, the first all-ceramic LC resonator for passive wireless temperature sensing was demonstrated up to 1200 °C. The work showed that the geometry of the LC resonator can be adjusted to accommodate the semiconducting ITO electrodes where the electrical conductivity can be as low as 90 S/cm. The LC resonator was fabricated on an Al_2_O_3_ substrate via screen-printing technology and co-sintered at 1250 °C for 4 h. The microstructural investigation of the parallel plate capacitor and the inductor showed that ITO and the Al_2_O_3_ grains were well percolated in all three dimensions with an average grain size of about 2.20 µm. The cross-sectional micrograph reveals that there are no interfacial defects or chemical reactions at the ITO–Al_2_O_3_ interface. The individual circuit elements (parallel plate capacitor and the inductor) were evaluated from room temperature to 1200 °C to study the temperature-dependent relation. The Δ*L* from 150 to 1200 °C was measured to be ~5 nH since it does not depend on the change in dielectric permittivity. However, the Δ*C* increased with the temperature, proving that the temperature sensing is purely due to Δ*C* as a function of temperature. The thermal loading cycle (heating and cooling) revealed that the capacitor demonstrated a hysteresis of ~1.39% between 150 and 1200 °C.

The resonant frequency (*f_r_*) of the LC resonator was characterized from 200 to 1200 °C for both the heating and cooling cycle. The experimental Δ*f_r_* as a function of temperature from 200–1200 °C agrees with the computational simulation. The Δ*f_r_* as a function of temperature was repeatable and followed the same trend during the thermal loading cycle. The LC resonator developed in this work shows a temperature sensitivity of 170 kHz/°C from 200–1200 °C. It is significant to note that the quality factor (*Q_f_*) of the LC resonator does not attenuate at elevated temperatures due to the increase in conductivity of the ITO. The increase in *Q_f_* is contrary to the use of metallic electrodes, which deteriorates at elevated temperatures (>1000 °C), leading to diminished sensitivity and reliability. Based on the evaluations presented in this work, stable semiconducting high-temperature ceramics are a viable alternative to metallic electrodes for harsh-environment passive wireless sensing applications. The future work will focus on evaluating various electronically conductive ceramic systems for high-temperature passive wireless applications.

## Figures and Tables

**Figure 1 sensors-22-02165-f001:**
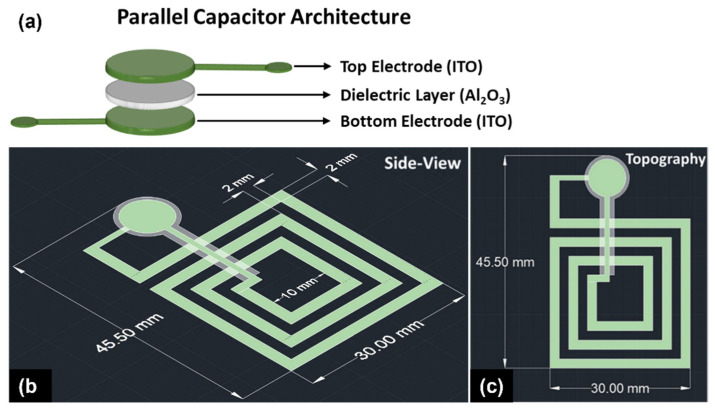
Schematic representation of the (**a**) parallel capacitor architecture where the diameter of the Al_2_O_3_ dielectric layer was kept 10% larger than the top and the bottom electrodes to prevent short-circuiting the capacitor, (**b**) passive wireless LC resonator consisting of parallel capacitor and planar inductor, and (**c**) topography of the multi-layered stack.

**Figure 2 sensors-22-02165-f002:**
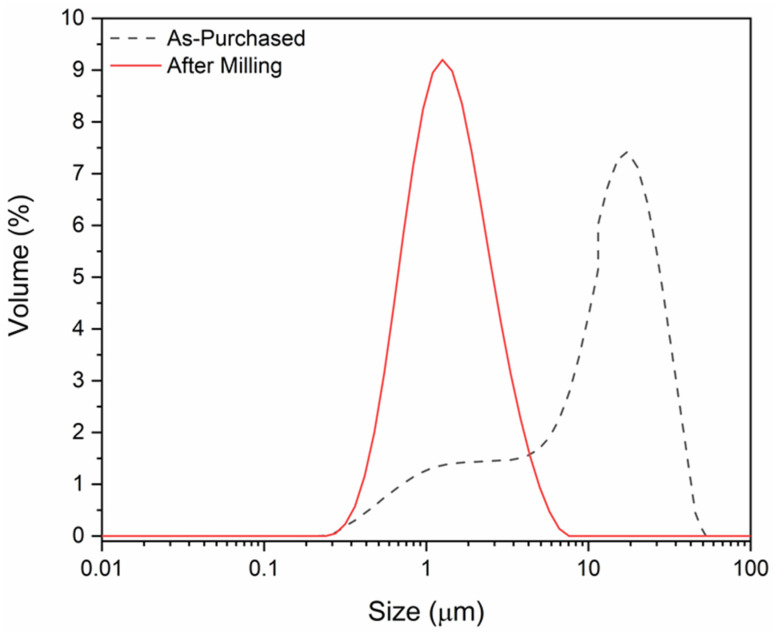
Comparison of the particle size distribution of the as-purchased and the attrition-milled ITO.

**Figure 3 sensors-22-02165-f003:**
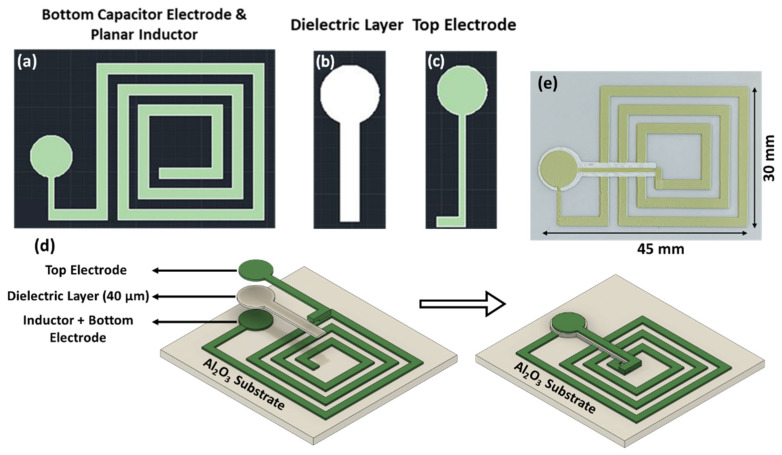
Schematic of proposed LC resonator consisting of (**a**) bottom capacitor electrode and planar inductor; (**b**) dielectric layer with the insulation bridge; (**c**) top electrode connecting the inner inductor to the top capacitor to complete the LC circuit; (**d**) 3D schematics of the different layers of the LC resonator; (**e**) optical microscope image of the LC resonator after co-sintering at 1250 °C for 4 h in ambient atmospheric conditions.

**Figure 4 sensors-22-02165-f004:**
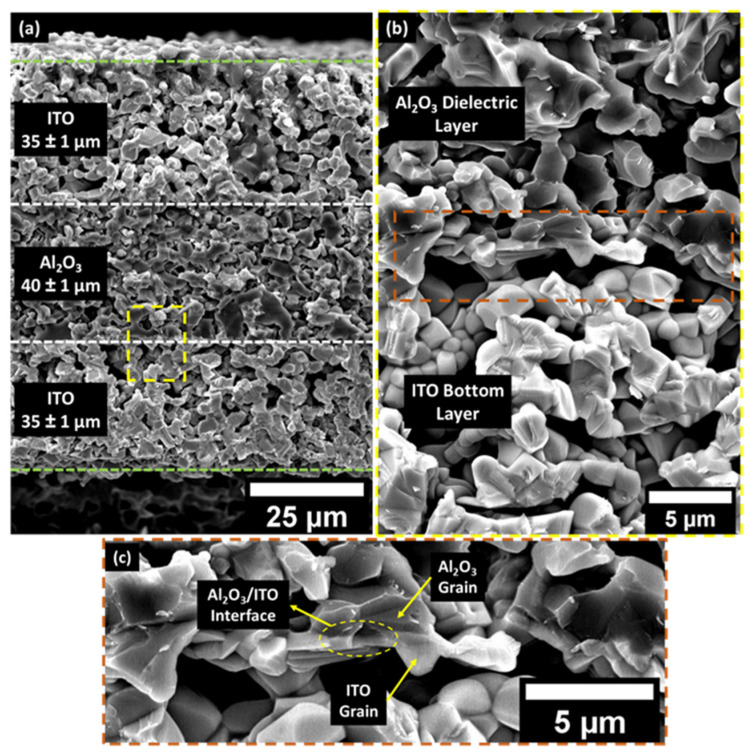
SEM micrograph of the cross-section of the parallel capacitor showing (**a**) all three layers of the ITO–Al_2_O_3_–ITO co-sintered at 1250 °C. The area between two white dotted lines represents the Al_2_O_3_ dielectric layer, whereas the area between the white and green dotted lines represents ITO top and bottom electrodes. (**b**) Magnified image of the ITO–Al_2_O_3_ dielectric layer (yellow dotted box in (**a**)) showing that there is a clear boundary between the interfaces and no microscopic reaction at the ITO–Al_2_O_3_ interface. (**c**) High magnification image showing the percolation of ITO and Al_2_O_3_ grains (dark orange box in (**b**)).

**Figure 5 sensors-22-02165-f005:**
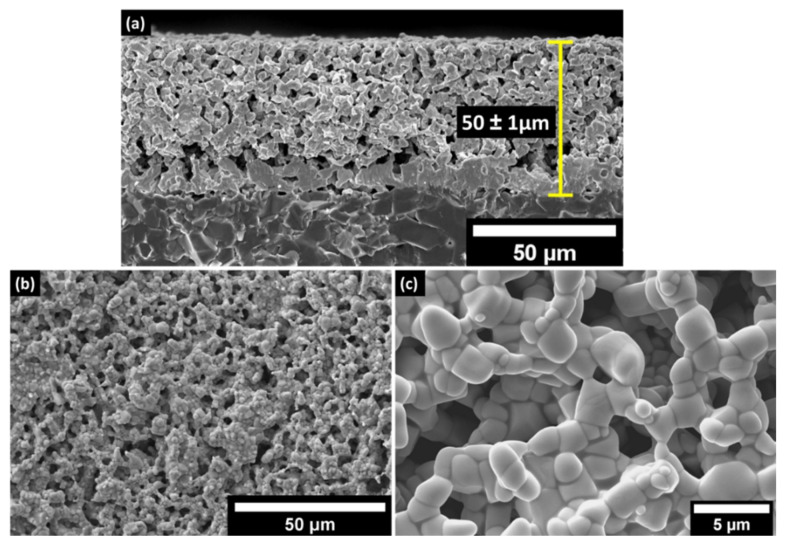
SEM micrograph of the (**a**) cross-section of the planar inductor deposited on Al_2_O_3_ substrate sintered at 1250 °C showing that the thickness of the ITO layer is 50 μm; (**b**,**c**) topography of the ITO electrodes and the magnified image revealing the percolated grains of ITO with an average grain size of 2.2 ± 0.62 μm.

**Figure 6 sensors-22-02165-f006:**
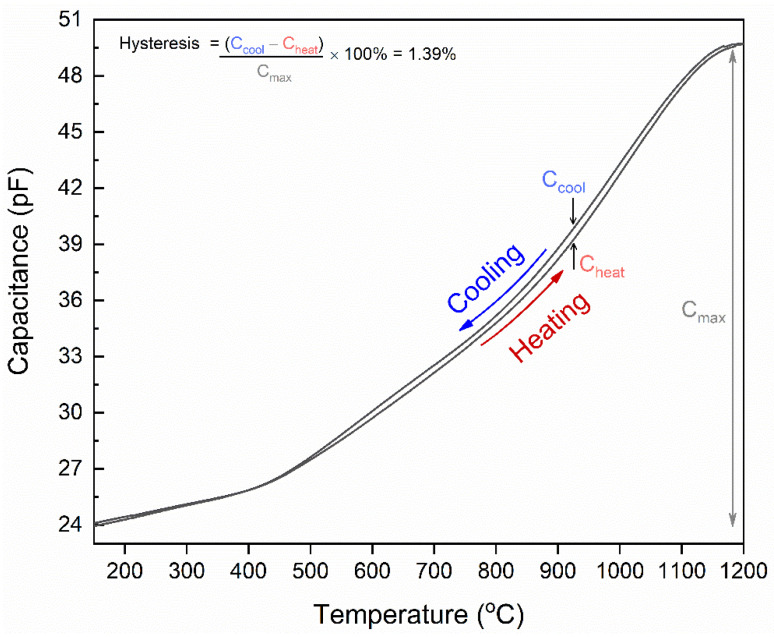
The temperature-dependent capacitance of the parallel capacitor from 100–1200 °C in ambient atmospheric conditions.

**Figure 7 sensors-22-02165-f007:**
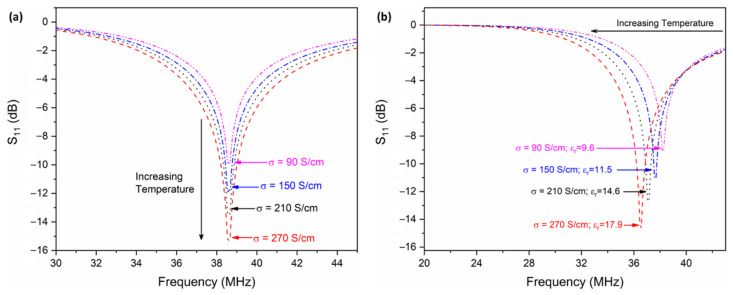
Wireless response of the LC resonator simulated by (**a**) varying only the electrical conductivity of ITO from room temperature (~90 S/cm) to 1200 °C (270 S/cm); (**b**) changing both the electrical conductivity of ITO and the dielectric permittivity of Al_2_O_3_ layer. The permittivity was calculated from capacitance versus temperature using Equation (3), assuming no physical change in the dimension due to temperature.

**Figure 8 sensors-22-02165-f008:**
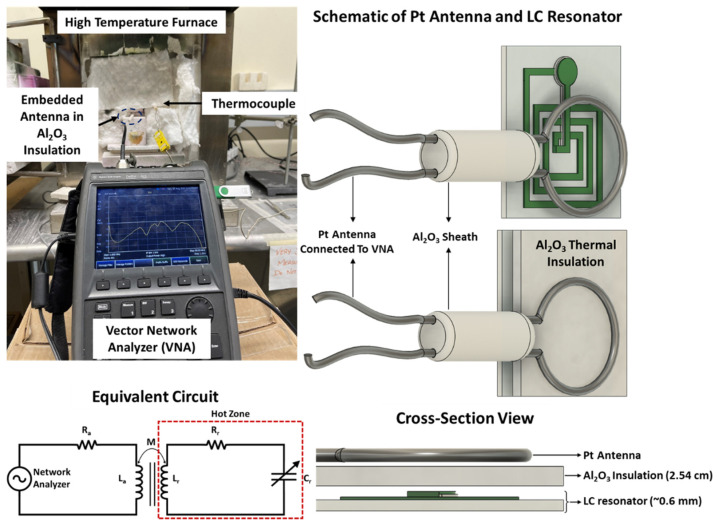
Picture of the experimental setup used to characterize the wireless response as a function of temperature. Schematic of the LC resonator characterization depicts the Pt antenna thermally shielded from the hot zone by a 1-inch (~2.54 cm)-thick Al_2_O_3_ insulation. The equivalent circuit diagram is also shown, which presents the principle of mutual inductive coupling between the reader antenna and the LC resonator in the hot zone.

**Figure 9 sensors-22-02165-f009:**
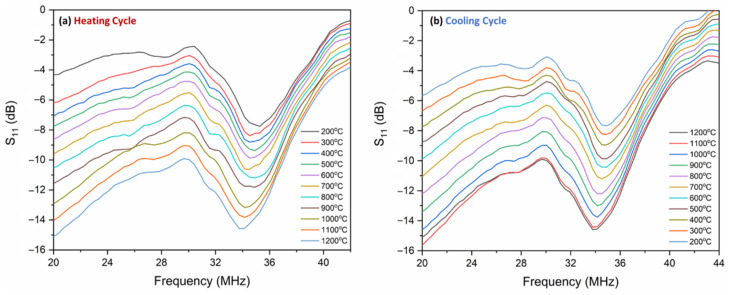
Wireless response of the LC resonator during (**a**) the heating and (**b**) the cooling cycle with a steady rate of 1 °C/min. The shift in the resonant frequency curve is due to temperature variation.

**Figure 10 sensors-22-02165-f010:**
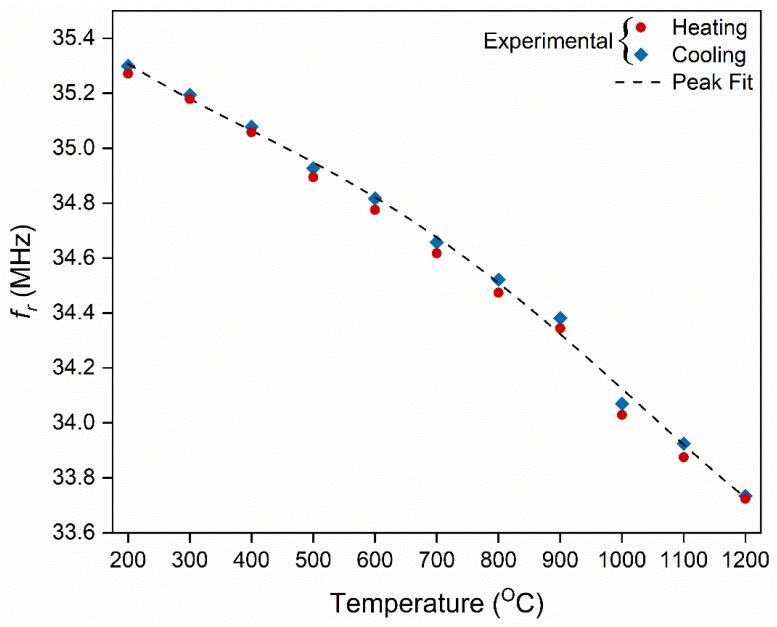
Extrapolated resonant frequency minima of the heating and cooling cycle of the LC resonator. The peak fit is a third-order nonlinear polynomial function.

**Figure 11 sensors-22-02165-f011:**
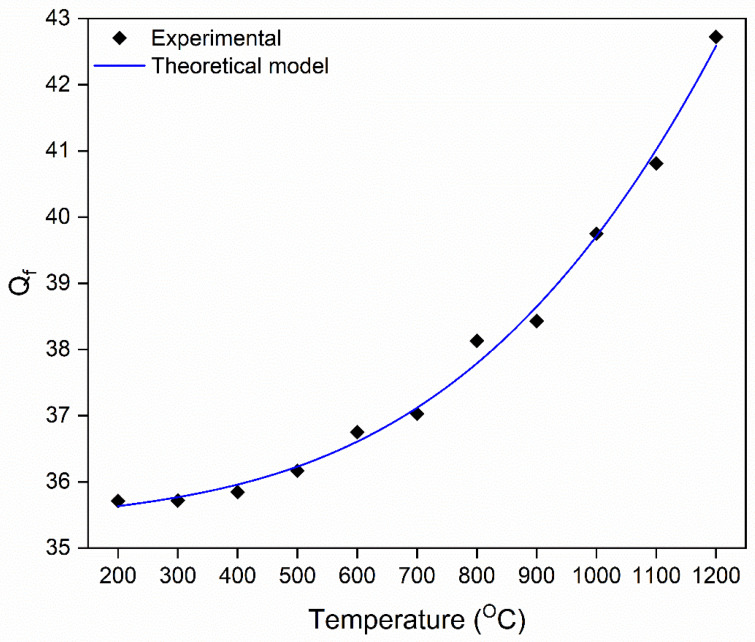
The temperature-dependent quality factor (*Q_f_*) of the parallel capacitor from 100 °C to 1200 °C in ambient atmospheric conditions.

**Table 1 sensors-22-02165-t001:** Design parameters of the parallel plate capacitor and the square planar inductor.

Parallel Plate Capacitor (*C_r_*)		Planar Inductor (*L_r_*)	
ITO electrode diameter (mm)	3.51	Electrode width (mm)	2
Al_2_O_3_ dielectric diameter (mm)	3.86	Spacing b/w turns (mm)	
Electrode area (mm^2^)	9.68	Number of turns	3
*ε_r_* (Al_2_O_3_ @25 °C)	9.6	Inner diameter (mm)	10
Al_2_O_3_ dielectric thickness (µm)	40	Outer diameter (mm)	30

## Data Availability

Not applicable.

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
