# Peer review of "All-Ceramic Passive Wireless Temperature Sensor Realized by Tin-Doped Indium Oxide (ITO) Electrodes for Harsh Environment Applications"

_sensors, 2022, doi:10.3390/s22062165_

Round 1
Reviewer 1 Report
This paper presents an all-ceramic passive wireless LC resonator for high temperature sensing up to 1200℃. The concept of this paper is promising, and the paper is well written. But it still contains some problems. Here are some questions and suggestions:
- It is mentioned in the title, abstract and introduction that the ITO electrode is doped with SnO2. However, the reviewer did not see any doping process in section 2, and the improvement of the Qf after doping SnO2 did not show in the paper.
- In the abstract, the author should highlight the innovation of electrode materials and the improvement of LC resonator performance instead of introducing the detailed structure or the preparation process. Besides, in line 21 and 22, the author stated“The sensitivity (average change in resonant frequency with respect to temperature) from 200-1200℃ was ~74.56 kHz/℃”. However, the author only presented the experimental results in figure 10, but did not calculate the sensitivity of the resonator.
- In the introduction, the author mentioned that the signal loss of the reported LC resonators is too high, but does the new type of electrode material solve this problem? The reviewer suggests the author increase the demonstration in this part. It would be better if the author could give the experimental evidence of decreasing the signal loss in this work.
- In line 117, the thermal expansion coefficient should be 10-6/o
- The C-T curve in figure 6 is plotted through 3rd order polynomial fit, but it does not show original data points in the photograph. The review suggests the author add them.
- In section 3.3, the author simulated the electrical properties of the LC resonator by ANSYS HFSS. What are the structural parameters of the LC resonator used in the simulation? Are they the same as the designed parameters? Is there a diagram of the LC resonator established in the simulation software?
- In line 441 and 442, the author states “The experimental fr of the LC resonator at room temperature was measured to be 35.43 MHz which is 3.1 MHz lower than the theoretical value”. However, the experimental results shown in figure 9 and figure 10 are in the temperature range of 200-1200℃ without any room temperature results.
In general, the reviewer suggests the author put more efforts on this manuscript and revise it carefully.

Author Response
Dear Reviewer,
The paper entitled, "All-Ceramic Passive Wireless Temperature Sensor Realized by Tin-doped Indium Oxide (ITO) Electrodes for Harsh Environment Applications" was revised based on your valuable comments. All the changes to the manuscript were completed. We have listed our point-by-point response to your comments in the attached document. The authors appreciate considering our manuscript for publication.

Reviewer 2 Report
Dear Author(s),
I ask you to make some improvements and some corrections of some mistakes and details as outlined below.
General remarks:
- The unit for temperature is usually written with the space between value and unit (25 °C). It is your and Editor’s decision to correct or not.
- In the manuscripts, the word Figure is usually written with the capital letter F.
Asking for improvements:
- Please check the title. In my opinion, it is a little bit confused.
- Please check the calculated values in Lines 172, 173. The capacitance value does not correspond to the thickness of the ITO.
- Please verify the calculated values in Lines 374, 375, where is written: “The average change in capacitance (ΔC) is ~26 pF from 100 – 1200°C with a unit average of 0.234 pF/°C”.
- Please give the calculated and measured values capacitance and inductance at room temperature
- Please verify the thickness of ceramic “thermal shield” Lines 423 and 438 and in Figure 8.
Remarks on References:
- References 5 and 25 are the same (Lines 580, 624)
- Please format References 29 and 30 (Lines 632, 633)
- I cannot verify Reference 30
- I didn’t check all References, but I am afraid that some References have not been used correctly. Just a few examples, the References in Lines: 67, 85, 87, maybe more,
- Please give the Reference for the sentences in Lines 81-83
- If it is possible, please give an additional Reference (reference 41 is self-citation) for the value (90 S/cm) of the electrical conductivity of ITO electrodes.
Please verify small remarks:
Line 36: low-temperature
Lines 36, 37: Conventional Resistance Temperature Detectors (RTDs) and thermocouples …..
Line 38: Thermocouples and thermistors have limited applications
Line 44: and lead to the necessary frequency calibration
Line 45: sensing temperature
Line 51: The conventional material systems
Line 56: high-temperature
Line 84: onto
Line 106: high-temperature
Lines 132, 133: to corroborate the stability of the wireless response at high temperatures.
Line 235: The screen-printed
Line 255: were
Line 356: temperature-dependent
Line 363: temperature-dependent
Line 383: The temperature-dependent
Line 393-394: 800°C, 1200°C)
Line 404: 800°C, 1200°C)
Line 426: high-temperature
Line 474: The fr follows a similar trend during the cooling and heating cycle.
Line 523: Figure 11. The temperature-dependent quality factor (Qf) of the parallel capacitor from 100°C to 1200°C in ambient atmospheric conditions.
Line 533: grain size of about 2 μm.
Line 561: the US Department
Line 564: the US Department
Author Response

(The authors gave the same response as above.)

Round 2
Reviewer 1 Report
Accept in present form